# Monitoring of Unfractionated Heparin Therapy in the Intensive Care Unit Using a Point-of-Care aPTT: A Comparative, Longitudinal Observational Study with Laboratory-Based aPTT and Anti-Xa Activity Measurement

**DOI:** 10.3390/jcm11051338

**Published:** 2022-02-28

**Authors:** Benjamin Lardinois, Michaël Hardy, Isabelle Michaux, Geoffrey Horlait, Thomas Rotens, Hugues Jacqmin, Sarah Lessire, Pierre Bulpa, Alain Dive, François Mullier

**Affiliations:** 1Hematology Laboratory, CHU UCL Namur, Université Catholique de Louvain, 5530 Yvoir, Belgium; michael.hardy@uclouvain.be (M.H.); hugues.jacqmin@uclouvain.be (H.J.); francois.mullier@uclouvain.be (F.M.); 2Anesthesiology Department, CHU UCL Namur, Université Catholique de Louvain, 5530 Yvoir, Belgium; sarah.lessire@uclouvain.be; 3Namur Thrombosis and Hemostasis Center (NTHC), CHU UCL Namur, Université Catholique de Louvain, 5530 Yvoir, Belgium; 4Department of Intensive Care, CHU UCL Namur, Université Catholique de Louvain, 5530 Yvoir, Belgium; isabelle.michaux@uclouvain.be (I.M.); geoffrey.horlait@uclouvain.be (G.H.); thomas.rotens@uclouvain.be (T.R.); pierre.bulpa@uclouvain.be (P.B.); alain.dive@uclouvain.be (A.D.)

**Keywords:** heparin, monitoring, unfractionated heparin, APTT, POCT, anti-Xa

## Abstract

Continuous intravenous unfractionated heparin (UFH) is administered routinely in the intensive care unit (ICU) for the anticoagulation of patients, and monitoring is performed by the activated partial thromboplastin time (APTT) or anti-Xa activity. However, these strategies are associated with potentially large time intervals before dose adjustments, which could be detrimental to the patient. The aim of the study was to compare a point-of-care (POCT) version of the APTT to (i) laboratory-based APTT and (ii) measurements of anti-Xa activity in terms of correlation, agreement and turnaround time (TAT). Thirty-five ICU patients requiring UFH therapy were prospectively included and followed longitudinally for a maximum duration of 15 days. UFH was administered according to a local adaptation of Raschke and Amanzadeh’s aPTT nomograms. Simultaneous measurements of POCT-APTT (CoaguCheck^®^ aPTT Test, Roche Diagnostics) on a drop of fresh whole blood, laboratory-based APTT (C.K. Prest^®^, Stago) and anti-Xa activity (STA^®^Liquid anti-Xa, Stago) were systematically performed two to six times a day. Antithrombin, C-reactive protein, fibrinogen, factor VIII and lupus anticoagulant were measured. The time tracking of sampling and analysis was recorded. The overall correlation between POCT-APTT and laboratory APTT (*n* = 795 pairs) was strongly positive (rs = 0.77, *p* < 0.0001), and between POCT-APTT and anti-Xa activity (*n* = 729 pairs) was weakly positive (rs = 0.46, *p* < 0.0001). Inter-method agreement (Cohen’s kappa (k)) between POCT and laboratory APTT was 0.27, and between POCT and anti-Xa activity was 0.30. The median TATs from sample collection to the lab delivery of results for lab-APTT and anti-Xa were 50.9 min (interquartile range (IQR), 38.4–69.1) and 66.3 min (IQR, 49.0–91.8), respectively, while the POCT delivered results in less than 5 min (*p* < 0.0001). Although the use of the POCT-APTT device significantly reduced the time to results, the results obtained were poorly consistent with those obtained by lab-APTT or anti-Xa activity, and therefore it should not be used with the nomograms developed for lab-APTT.

## 1. Introduction

Despite the advent of low molecular weight heparins (LMWHs), continuous intravenous (IV) unfractionated heparin (UFH) anticoagulation remains useful in the intensive care unit (ICU) in several indications, such as circulatory assist devices, severe renal failure and high bleeding risk. 

Patients treated with UFH are commonly monitored by the activated partial thromboplastin time (APTT) [1]. This coagulometric assay measures the time required to form a fibrin clot after contact activation of platelet-poor plasma [2], and is therefore an indirect assessment of heparin activity [3]. However, especially in critically ill patients [4], APTT monitoring has many drawbacks, including a lack of clinical validation of the therapeutic range (i.e., prolongation of 1.5 to 2.5 times baseline APTT) [1] derived from the work of Basu et al. in 1972 [5], the lack of international standardization [4], the presence of a high technical bias [6,7] and many confounding factors affecting this test (e.g., low fibrinogen or coagulation factor levels, high factor VIII levels, presence of a lupus anticoagulant (LA), preactivation of samples during difficult collection). In this context, monitoring and dose titration of IV UFH by chromogenic anti-Xa activity measurement has been suggested as the preferred method, given its more specific assessment of heparin levels and its independence from inflammatory factors. However, the commonly accepted therapeutic range of 0.3–0.7 IU/mL of this assay is not yet clinically validated and shows wide variations between laboratories [8].

In 2012, the American College of Chest Physicians called for the identification of an optimal approach to UFH monitoring [1], but there is still no consensus regarding the optimal anticoagulation strategy and its monitoring, even in specific contexts such as extracorporeal membrane oxygenation (ECMO) [2,9].

Regardless the method used, in clinical practices there are difficulties in monitoring UFH treatment. For example, there is often a delay between blood collection and UFH dose adjustment due to the time required for blood sample transport to the central laboratory, for processing of the sample including centrifugation, and for analysis and communication of the results. This may increase the time to therapeutic anticoagulation and therefore could be detrimental to the patient. In this context, Roche Diagnostics has developed a new APTT reagent (CoaguCheck^®^ aPTT Test) on its point-of-care (POCT) system (i.e., CoaguChek^®^ Pro II). However, the CoaguCheck^®^ aPTT still lacks clinical validation in patients undergoing IV UFH therapy.

The aim of the study was to compare this POCT version of the APTT to laboratory-based APTT and measurements of anti-Xa activity in terms of correlation, agreement, relationship and turnaround time (TAT) in patients under UFH therapy in the ICU.

## 2. Methods

We conducted a monocentric, observational, longitudinal and prospective study from July 2019 to July 2020 by comparing the point-of-care CoaguCheck^®^ aPTT Test (Roche Diagnostics) to (i) laboratory-based APTT and (ii) anti-Xa activity.

### 2.1. Patient Selection

Adult patients requiring UFH therapy for at least 72 h between July 2019 and July 2020 were included. Patients with a previous anticoagulation, thrombolytic therapy or any contraindication to heparin therapy, such as heparin-induced thrombocytopenia (HIT) at the time of UFH therapy initiation, were excluded from the study. We also excluded COVID-19 patients from the analysis. Discharge from the ICU, completion of end-organ support therapy (continuous veno-venous hemofiltration (CVVH), ECMO), or a maximum duration of 15 days marked the end of their inclusion. Due to the exploratory design of the study, no formal sample size calculation could be performed. The study was approved by the Ethics Committee of the CHU UCL Namur (Belgium accession number B039201940886). Informed consent was obtained from the patient or his/her legal representative in case of altered consciousness.

### 2.2. UFH Management

In brief, UFH was administered according to a local adaptation of Raschke [10] and Amanzadeh’s [11] aPTT nomograms, depending on weight, clinical indication, anticipated risk of bleeding and patient’s baseline lab-aPTT before any anticoagulation (Table 1). UFH doses were adjusted on lab-APTT for low range (1.5–2.0 times baseline APTT) or high range (2.0–2.5 times baseline APTT) anticoagulation by monitoring the aPTT every 4 to 12 h according to the local protocol (depending on the stability of the aPTT between consecutive measurements). The clinical outcomes, including complications, such as bleedings or thromboembolism, were also recorded for each patient.

### 2.3. POCT-APTT, Laboratory-Based APTT and Anti-Xa Activity

Simultaneous measurements of POCT-APTT (CoaguCheck^®^ aPTT Test, Roche Diagnostics, Basel, Switzerland) on a drop of fresh whole blood using the CoaguChek^®^ Pro II device (Roche Diagnostics), laboratory-based APTT (lab-APTT; C.K. Prest^®^, Stago, Asnières-sur-Seine, France) and anti-Xa activity (STA^®^Liquid anti-Xa, Stago, Asnières-sur-Seine, France) on 109 mM buffered citrated platelet-poor plasma (Vacuette^®^, Greiner Bio-One, Courtaboeuf, France) after a double centrifugation for 15 min at 1500× *g* at 20 °C, were systematically performed before dose adjustment. Lab-APTT uses kaolin as activator and cephalin from rabbit brains. The CoaguCheck^®^ aPTT Test is a one-step coagulation test using celite as an activator and a mixture of defined phospholipids. Each test strip has a test area containing an aPTT reagent. When blood is applied, the reagent is dissolved, and an electrochemical reaction takes place, which is transformed into a clotting time value. The clotting time value is automatically converted into plasma-equivalent seconds according to a calibration based on the laboratory APTT method actin FSL (Siemens) [12]. STA^®^ Liquid anti-Xa is a chromogenic assay measuring the residual activity of added factor Xa after inhibition by antithrombin potentiated by heparin administered to the patient. The test uses endogenous antithrombin (AT), does not contain dextran sulfate and is calibrated with UFH standards. 

Both lab-aPTT and anti-Xa activity were performed on the STA-R MAX2 analyzer (Diagnostica Stago, Asnières-sur-Seine, France). The intra- and inter-assay coefficients of variation observed from the manufacturer’s pathological commercial controls were: 2.1% and 5.1% for POCT-APTT (according to the manufacturer), 0.7% and 1.7% for lab-APTT, and 3.3% and 4.4% for anti-Xa activity, respectively. The lower limit of quantification of POCT-APTT, lab-APTT and anti-Xa activity according to the manufacturer was 20 s, 20 s and 0.1 IU/mL, respectively.

The usual therapeutic range for the lab-APTT ratio is 1.5–2.5, as defined by Basu et al. in 1972 [5]. This therapeutic range was previously confirmed for C.K. Prest^®^ [13]. We divided the lab-APTT ratio into low (1.5–2.0) and high (2.0–2.5) therapeutic ranges according to the different type of anticoagulation indications (e.g., CVVH: 1.5–2.0; venous thromboembolism or mechanical heart valve: 2.0–2.5). The same target ranges were used for the POCT-APTT ratio as specified by the manufacturer. For anti-Xa activity, therapeutic heparin levels (i.e., 0.3 to 0.7 IU/mL) [14,15] were further split into low a therapeutic range (0.3–0.5 IU/mL) and a high therapeutic range (0.5–0.7 IU/mL).

The POCT-APTT ratio was obtained by the following formula:$$\mathrm{POCTAPTT}\;\mathrm{ratio}=\frac{\mathrm{POCTAPTT}\;\mathrm{under}\;\mathrm{anticoagulation}}{\mathrm{POCTAPTT}\;\mathrm{before}\;\mathrm{any}\;\mathrm{anticoagulation}}$$
where POCT-APTT under anticoagulation is the measurement after initiation of UFH therapy, variable at each sampling, and POCT-APTT before any anticoagulation is baseline POCT-APTT, unique for each patient. The same was realized for lab-APTT. We therefore obtained two pairs (POCT-APTT ratio:lab-APTT ratio and POCT-APTT ratio: anti-Xa) for each simultaneous measurement. Each pair was then assigned to one of the two categories (agreement, disagreement) according to their agreement with the infra-, normo- or supra-therapeutic range (Table 2), as suggested by Ratano et al. [8]. The disagreement category was further divided into two subcategories as follows: POCT ratio target range differing only from one target range (unsatisfactory) or differing from two target ranges (contradictory) of the lab-APTT ratio or the anti-Xa activity. For instance, a supra-therapeutic POCT ratio paired with an infra-therapeutic lab-APTT ratio was categorized as a disagreement, and subcategorized as contradictory.

### 2.4. Confounding Factors

The following additional laboratory tests were performed in parallel with POCT-APTT, lab-APTT and anti-Xa assays: antithrombin (AT; Liatest ATIII reagent, Stago, Asnières-sur-Seine, France), C-reactive protein (CRP; CRP Gold Latex^®^, DiAgam, Ghislenghien, Belgium), fibrinogen (Fg; STA Liquid Fib reagent, Stago, Asnières-sur-Seine, France), factor VIII (FVIII; STA Deficient VIII and STA-C.K. Prest reagents, Stago, Asnières-sur-Seine, France), factors XI and XII (FXI & FXII; STA Immunodef XI, STA Immunodef XII and STA-C.K. Prest reagents, Stago, Asnières-sur-Seine, France) and LA. For LA diagnosis, a full LA panel was performed twice in each patient (once in early ICU stay, once in late ICU stay) which comprised two screening tests (PTT-LA and STA-Staclot DRVV Screen, both reagents from Stago, Asnières-sur-Seine, France), mixing tests (50/50 *vol*/*vol* mixing with normal pooled plasma, same reagents) and confirmation tests (Staclot LA and STA-Staclot DRVV Confirm, both reagents from Stago, Asnières-sur-Seine, France). LA positivity was determined by two independent specialists in laboratory medicine based on the results of the whole panel and according to the guidance from the Scientific and Standardization Committee for lupus anticoagulant/antiphospholipid antibodies of the International Society of Thrombosis and Haemostasis [16]. Demographic variables including sex, age and weight were also collected from each patient.

### 2.5. Time Tracking

Time tracks of sampling, analysis, reading of results and dose modification were collected to calculate the different TATs. The time to the desired therapeutic target range, defined as the time-interval between the first injection of UFH and the achievement of one APTT measurement in the desired therapeutic range, was also collected for each patient.

### 2.6. Statistical Analysis

Unless otherwise specified, quantitative variables were expressed as the median (interquartile range; IQR) and categorical variables as a number (percent). The correlations between POCT-APTT and lab-APTT and anti-Xa activities were assessed by Spearman correlation coefficients (rs) after performing Wilcoxon matched-pairs signed-ranks tests. Statistical tests were two-tailed and results were considered statistically significant for α < 0.05. A correlation coefficient of 0.00–0.29, 0.30–0.49, 0.50–0.69, 0.70–0.89, or 0.90–1.00 was considered negligible, weak, moderate, strong, or very strong positive, respectively. Cohen’s kappa coefficients were used to calculate the inter-method agreements between the POCT-APTT ratio and the lab-APTT ratio, and between the POCT-APTT ratio and anti-Xa activity. The percentages of the different categories and subcategories described above were also calculated to appreciate the degree of agreement and disagreement. The correlation and the inter-method agreement between lab-APTT and anti-Xa activity were not evaluated in this study.

To identify variables influencing the relationship between POCT-APTT and lab-APTT, we fitted linear mixed-effects models (with variance components covariance structure) to account for intra-individual correlation. We used POCT-APTT as the dependent variable and lab-APTT as a fixed independent variable in all models. Other fixed independent variables considered in separate models were: CRP, fibrinogen, factor VIII, AT levels and positivity for LA; interaction terms with lab-APTT were also considered. Variables associated with the dependent variable with a *p*-value lower than 0.20 were retained in the initial multivariate model, which was simplified by backward elimination. *p*-values were estimated by F tests with denominator degrees of freedom calculated according to Kenward–Roger’s equation. R v.4.1.0 (Vienna, Austria) and GraphPad Prism 6.0e (California, CA, USA) were used to perform statistical analyses.

## 3. Results

### 3.1. Study Population

Among the 45 patients screened for eligibility, 35 were finally included in the study. For 29 of them (82.9%), a basal POCT-APTT measurement before UFH administration was available (Figure 1). Characteristics of the study population are summarized in Table 3. Among them, 23 (65.7%) required low range therapeutic anticoagulation for CVVH, deep vein thrombosis (DVT), ECMO or atrial fibrillation. Eight other patients (22.9%) received high range therapeutic anticoagulation for mechanical valves, pulmonary embolism (PE) or DVT. Four patients (11.4%) switched from the low to high therapeutic range of anticoagulation or conversely during the study period. The median population age was 64.7 years (IQR, 56.9–70.7). The median weight was 76.5 kgs (IQR, 65.0–94.5). The median duration of inclusion was 6.0 days (IQR, 5.0–11.2) and the median time to desired therapeutic target ranges was 29.1 h (IQR, 15.4–37.6). POCT-APTT was missing for 21.4 patient-days out of a total of 304 patient-days (7%). No thrombotic events were recorded.

### 3.2. POCT-APTT, Laboratory-Based APTT and Anti-Xa Activity

The obtained observations from the 29 patients (82.9%) with basal POCT-APTT and lab-APTT were used for the overall correlation, agreement and relationship ratio analyses. The six patients (17.1%) for whom basal POCT-APTT were not obtained were not included in the agreement and relationship ratio analyses, but their data were used for the overall comparison.

The overall correlations between POCT-APTT and laboratory APTT (*n* = 795 pairs), and POCT-APTT and anti-Xa (*n* = 729 pairs) are presented in Figure 2. The Spearman correlation coefficient (rs) observed for POCT vs. Lab-APTT (rs = 0.77, 95% confidence interval (CI), 0.74–0.79; *p* < 0.0001) was strongly positive and higher than the rs for POCT vs. anti-Xa (rs = 0.46, 95% CI, 0.40–0.52, *p* < 0.0001), which was weakly positive. The better correlation between POCT-APTT and lab-APTT was also observable in the single patient curves (Figure 3). After excluding the 17 patients on ECMO and CVVH (*n* = 18), we observed the following rs for POCT-APTT vs. lab-APTT and POCT-APTT vs. anti-Xa: 0.79 (95% CI, 0.75–0.82, *p* < 0.0001) and 0.59 (95% CI, 0.51–0.65, *p* < 0.0001), respectively.

Concerning the 29 patients with basal POCT-APTT and basal lab-APTT measurements, the rs were higher for POCT-APTT vs. Lab-APTT ratios than for the POCT-APTT ratios vs. anti-Xa within overall therapeutic ranges, even when considering low and high therapeutic ranges separately (Figure 4). Moreover, higher rs were observed for high therapeutic ranges than for low therapeutic ranges within both relationships. The best correlation obtained was therefore for the POCT-APTT ratio vs. lab-APTT ratio, which was strongly positive in high therapeutic ranges (rs = 0.74, 95% CI, 0.67–0.80, *p* < 0.0001), whereas the worst correlation observed was for the POCT-APTT ratio vs. anti-Xa, which was negligible in low therapeutic ranges (rs = 0.24, 95% CI, 0.13–0.33, *p* < 0.0001).

The inter-method agreements between the POCT ratio and lab-APTT ratio and the POCT ratio vs. anti-Xa activity showed only slight agreements for both low and high therapeutic targets, and were as follows: POCT-APTT ratios vs. lab-APTT ratios for low or high ranges, Kappa 0.20 (95% CI 0.14–0.26) and Kappa 0.13 (95% CI 0.06–0.21), respectively; and POCT-APTT ratios vs. anti-Xa activity for low or high ranges, Kappa 0.11 (95% CI 0.09–0.27) and Kappa 0.08 (95% CI 0.00–0.16), respectively (Figure 5).

### 3.3. Confounding Factors

In individual models, CRP, fibrinogen, factors VIII, XI and XII, AT levels and LA positivity modulated the relationship between POCT-APTT and lab-APTT (*p* = 0.005, *p* = 0.047, *p* = 0.005, *p* < 0.001, *p* < 0.001, *p* < 0.001 and *p* = 0.02, respectively; Appendix A); increasing CRP levels and LA positivity strengthened this relationship (increased correlation), whereas increasing fibrinogen, FVIII, FXI and FXII levels weakened it (decreased correlation); increasing AT levels and LA positivity also decreased the systematic difference between both tests, whereas FXI levels increased it. In the final multivariable model, CRP, fibrinogen, factor XII levels and LA positivity modulated the relationship between POCT-APTT and lab-APTT (Appendix A).

### 3.4. Time Tracking

The median turnaround times from sample collection to the lab delivery of results for lab-APTT and anti-Xa were 50.9 min (IQR, 38.4–69.1) and 66.3 min (IQR, 49.0–91.8), respectively (Mann–Whitney U 84473, two-tailed *p* value < 0.0001). Relative frequency distributions of these TATs are shown in Figure 6. These also represent the potential time savings of the POCT over the two laboratory measurements as it can be performed at the patient’s bed with a result obtained within 5 min. The median time from the delivery of the results on medical charts to its reading by the intensive care staff (day and night) was 32.4 min (IQR, 17.6–60.3); if required, heparin dose was adjusted within a mean of 3.6 min (sd, 14.2) after reading. The median global TAT from blood collection until dose adjustment based on laboratory APTT was 92.0 min (IQR, 69.3–121.2), regardless of the time, day or night.

## 4. Discussion

We investigated the comparison of a point-of-care version of the APTT to the lab-APTT assay and anti-Xa activity for monitoring UFH therapy in the ICU. To the best of our knowledge, this is the first longitudinal study evaluating the CoaguChek^®^ Pro II (Roche Diagnostics) with APTT reagent (CoaguCheck^®^ aPTT Test) in critically ill patients receiving UFH therapy.

Niederdöckl et al. first observed that the POC system showed reliable results for suspected coagulation factor deficiencies in a heterogeneous patient group [17]. Second, Arachchillage et al. evaluated this POCT device in 2019 on 80 ICU patients and suggested that the degree of anticoagulation in patients receiving UFH could not reliably be inferred from the POCT assay [4]. However, the majority of patients had been receiving ECMO and each patient was tested only once. In this longitudinal study, we observed that POCT-APTT measurements better correlated with lab-APTT than anti-Xa activity, especially in the high therapeutic range. The overall correlation was strong but not optimal for lab-APTT vs. POCT-APTT (rs = 0.77), and weak for anti-Xa vs. POCT-APTT (rs = 0.46), which is in accordance with this previous study [4]. This difference is mainly explained by the measurement principle used by the POCT, which is closer to lab-APTT than anti-Xa activity.

The measurement of APTT in heparin monitoring suffers from numerous drawbacks including interference from acute phase proteins, such as alpha-2-macroglobulin [18], or elevated concentrations of coagulation factors, such as factor VIII or fibrinogen [14,19]. Moreover, medical conditions encountered in intensive care patients, such as inflammation, infection or liver dysfunction, induce even more variations in these parameters, causing important discrepancies between APTT and heparin concentration [4]. Prolongation of APTT therefore does not necessarily correlate even more with heparin levels in patients with antithrombin deficiencies, and the impact of UFH therapy on APTT is age-dependent [20,21]. There is also a technical bias due to the variation in the sensitivity of APTT reagents to heparins [6,7], and a lack of international standardization of APTT [4], due to the numerous reagents and analyzers available on the market [22,23,24]. Given a measurement of aPTT in whole blood, the POCT device can also be influenced by several other factors independent of UFH doses, including platelet count and function or anemia [2,12,25]. Since anti-Xa activity is not affected by these factors and more specifically measures heparin activity [18], the correlation observed with the POCT was logically weaker. However this correlation was probably underestimated given that the lower limit of quantification of anti-Xa activity was 0.1 IU ml, and several POCT values were obtained for corresponding anti-Xa activities which were likely below this respective threshold.

In 2012, the American College of Chest Physicians had called for the identification of an optimal approach to UFH monitoring [1] due to the variable pharmacodynamics and pharmacokinetics of UFH. Anti-Xa activity was therefore incorporated by scientific societies such as the College of American Pathologists, the American College of Chest Physicians or laboratory guidelines such as the Clinical and Laboratory Standards Institute [26], which have recommended therapeutic ranges calibrated on the basis of anti-Xa chromogenic assays between 0.3 and 0.7 IU/mL, as suggested by Levine et al. [27]. However these therapeutic ranges show considerable variability between centers, are wide and not clinically validated [8]. In addition, analytical interferences on anti-Xa activity, including antithrombin deficiency (for the kits that do not add exogenous antithrombin to the reagents), hemolysis or hyperbilirubinemia due to the photo-optical detection method used, may also occur [9,28]. According to two previous studies, increased plasma-free hemoglobin concentrations resulted in a concentration-dependent underestimation of heparin activity [29,30]. As a consequence of shear stress and the exposure of blood to non-biological substances in ECMO patients, thrombocytopenia and altered platelet function may occur due to decreased levels of adhesion receptors, activation markers and surface expression of CD62/CD63 [9,31]. This may also underestimate UFH effects given measurements of anti-Xa activity in a platelet-free environment. Some patients may also experience high levels of fibrinogen during their ECMO course, to which anti-Xa assays are not sensitive. All of these conditions frequently encountered in patients on extracorporeal devices may therefore lead to inadequate UFH dosing [29,30,31,32]. Since 48.6% of the patients included in this study were on ECMO (8.6%) or CVVH (40.0%), this led to a poorer correlation between POCT-APTT and anti-Xa pairs. Factor II, in addition to FVIII, has been identified as a source of discrepancy between APTT and anti-Xa activity [33]. These discrepancies in aPTT and anti-Xa activity assays caused by between-method and between-laboratory variations still require convincing evidence regarding the use of anti-Xa-based therapeutic ranges and the calibration of APTT to anti-Xa. Although the number of institutions using anti-Xa for monitoring and dose titration is steadily increasing [34], studies comparing APTT-based protocols vs. anti-Xa-based protocols to monitor UFH infusion are scarce. Two comparative studies showed better therapeutic control when utilizing an anti-Xa protocol, considering earlier achievement of the therapeutic level and lower heparin dose requirement [35,36]. Another study suggested to use anti-Xa assays considering a lower accuracy for APTT to detect UFH underdosing and overdosing in critically ill patients [37].

Inter-method agreements observed were only fair for both relationships if considering overall therapeutic ranges and slight if considering low or high therapeutic ranges. The wider dispersion of values observed between the POCT-APTT and anti-Xa techniques reflects the lower correlation.

The percentages of pairs in agreement were higher for the POCT-APTT ratio vs. anti-Xa than vs. lab-APTT. However, Cohen’s Kappa coefficients were lower for low than high therapeutic ranges for this relationship. This discrepancy is probably due to the majority of POCT-APTT ratio vs. anti-Xa pairs falling in the anti-Xa and POCT-APTT ratio infra-therapeutic ranges. In addition, POCT-APTT ratios were, most of the time, lower than lab-APTT ratios in the disagreement categories. This underestimation of the POCT-APTT ratio compared to the other two tests was observable by a left shift on the correlation graphs (Figure 4) and indicate the existence of a bias. This also suggests that the therapeutic cut-offs (1.5 and 2.5) of the POCT measurement specified by the manufacturer may be inadequate. These cut-offs are generally dependent on the measurement principle and could not be transposable to other techniques. Therapeutic ranges specific to POCT-APTT should therefore be determined and validated. Moreover, the therapeutic range of 1.5–2.5 times the control APTT is still uncertain [38], and is known to have introduced more varied clinical decisions concerning UFH therapy [26]. This is also reinforced by clinical observations of hemorrhagic events and the absence of thrombotic events, while the values of anti-Xa and lab-APTT were mostly subtherapeutic.

In light of these considerations, lab-APTT-based heparin dosing nomograms should not be used with the POCT-APTT assay, which is consistent with previous observations [4].

We also assessed the variables that could explain differences between POCT-APTT and lab-APTT and found that CRP, fibrinogen, factor XII levels, and LA positivity could explain differences between the two tests. Recent studies found that FXII and FXIII show a significant decrease during ECMO, FXIII even showing reduced levels hours before its initiation [39,40]. Low coagulation FXII activity has been associated with less thromboembolic complications [40], and was beneficial in an animal model, as its inhibition prevented fibrin deposition and thrombosis in the extracorporeal circuit [41]. As the whole cohort in our study included patients under the extracorporeal circuit, we assumed it could influence the correlations between POCT-APTT and anti-Xa assays and POCT-APTT vs. lab-APTT.

This study also demonstrated the time savings of POCT compared to the other two laboratory measurements. The lab-APTT and anti-Xa techniques took a median of 50.9 min and 66.3 min to obtain results, respectively, while POCT measurements required less than 5 min from finger prick test to available results, which is its main advantage. This reduction of the time between blood collection and dose adjustment could be associated with a benefit in the management of critically ill patients, for example by minimizing the time below or above therapeutic range, which could help reduce thrombotic and hemorrhagic complications, respectively. However, this should be evaluated in dedicated prospective studies.

As previously observed in another study [18], heparin therapy was mostly sub-therapeutic, notably because reaching the therapeutic range required a median delay of 29.1 h (IQR, 15.4–37.6), but also because some clinicians are reluctant to administer excessive anticoagulants because of the fear of bleeding complications. Appendix A summarizes the characteristics of each test. 

The strength of this analytical evaluation lies in the high number of paired measurements among a heterogenous group of critically ill patients requiring different indications of heparin therapy, representing daily intensive care practice.

This monocentric study has some limitations. First, the number of patients was limited, but each patient was followed longitudinally, making more than 700 test results available for analysis. Furthermore, an extensive longitudinal design is difficult to apply to a large number of patients. Second, there is huge analytical and biological variation in APTT response to UFH as in critically ill patients. Although we studied the effect of known confounding factors on APTT measurements and found that CRP, fibrinogen, factor XII levels and LA positivity were the principal factors that could modulate the relationship between POCT- and lab-APTT, there was a lot of between- and within-patient variability that was not related to UFH levels and that we could not explain with our model [38]. In addition, the inter-operator variability for POCT measurements could not be minimized due to the day and night conduct of the study. Third, patient monitoring was based on lab-APTT and not compared to a cohort of patients monitored by anti-Xa activity or POCT-APTT; thus, it has not been possible to assess the effectiveness of anticoagulation based on these two assays. Fourth, heparin monitoring was often sub-therapeutic according to our targets. More data in the supra-therapeutic ranges would be required. Fifth, basal POCT-APTTs were not obtained for 17.1% of the patients, which were forgotten due to heavy nursing workload.

Future studies should evaluate if POCT-APTT time savings could significantly increase the time in the heparin therapeutic range in comparison to lab-APTT and anti-Xa, and if this increase is associated with a reduced rate of thrombotic and/or haemorrhagic events. Clinical studies recording bleeding and thrombotic events with prospective cohorts of patients monitored by POCT compared to cohorts of patients monitored by anti-Xa or APTT are therefore necessary. In the meantime, UFH monitoring should be performed with anti-Xa activity or APTT from the laboratory.

## 5. Conclusions

Monitoring strategies based on anti-Xa activity or laboratory APTT are associated with potentially long intervals before dose adjustments, thus increasing the time required to obtain the therapeutic heparin range, which could be detrimental to the patient. In this context, the POCT-APTT device significantly reduces the time to results, but due to only slight agreements observed with lab-APTT and anti-Xa activity, its use could not be recommended when using activated partial thromboplastin time (aPTT)-based nomograms. Dedicated studies should, however, study UFH monitoring by POCT-APTT to assess whether the time saving in obtaining results translates into an increase in the time in the therapeutic range, a decrease in the incidence of complications, as well as to determine its optimal therapeutic range.

## Figures and Tables

**Figure 1 jcm-11-01338-f001:**
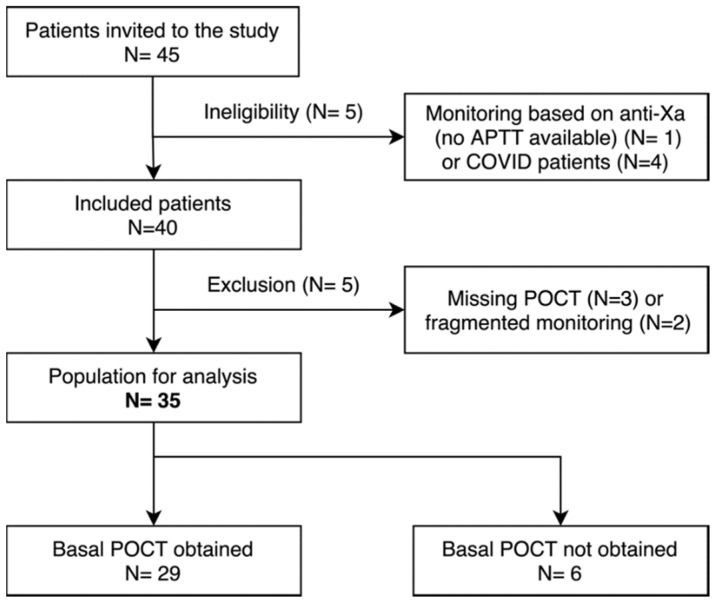
Flow diagram of identification, eligibility and inclusion processes. The basal POCT is the first POCT obtained before starting UFH administration. COVID, coronavirus disease 2019; POCT, point of care test; UFH, unfractionated heparin.

**Figure 2 jcm-11-01338-f002:**
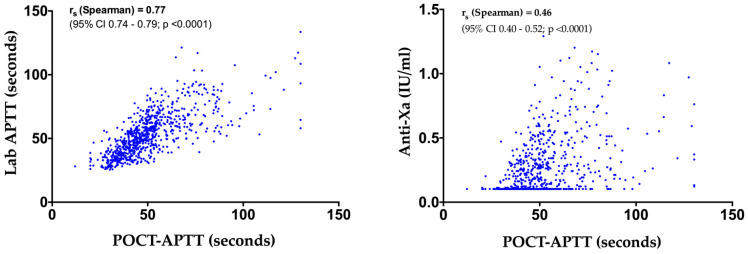
Overall correlation of POCT-APTT vs. laboratory APTT (**left** panel, *n* = 795) and POCT-APTT vs. anti-Xa (**right** panel, *n* = 729) from the 35 patients included in the study.

**Figure 3 jcm-11-01338-f003:**
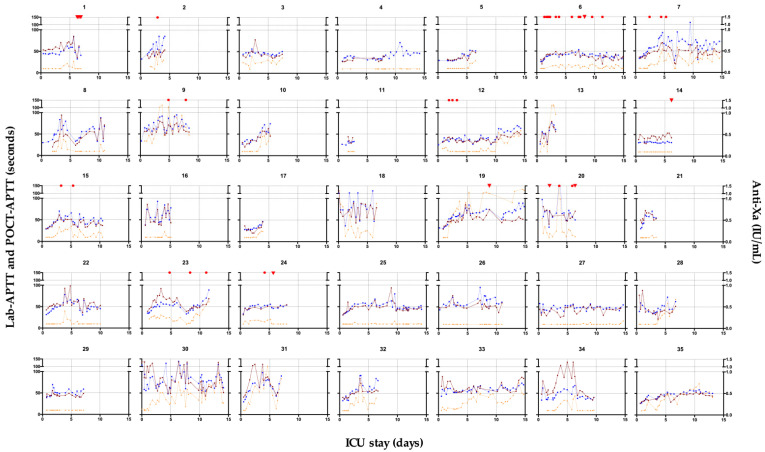
Temporal changes in POCT-APTT (brown), lab-APTT (blue) and anti-Xa (orange) levels during ICU stays for the 35 patients. Major or minor bleedings are symbolized by red triangles or dots above the graphs, respectively. No thrombotic events were recorded. APTT, activated partial thromboplastin time; ICU, intensive care unit; POCT, point-of-care testing.

**Figure 4 jcm-11-01338-f004:**
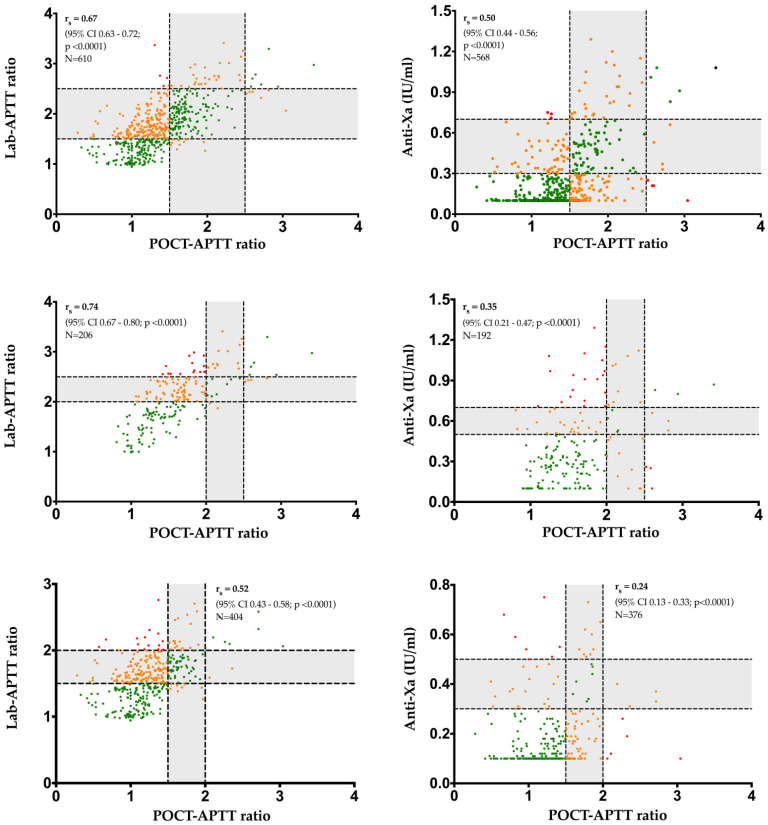
Correlations of the POCT-APTT ratios vs. laboratory APTT ratios (left panels) and POCT-APTT ratios vs. chromogenic anti-Xa activity measurement (right panels), according to overall (top panels), high (center panels) or low (bottom panels) therapeutic target ranges from the 29 patients with basal POCT and basal lab-APTT measurements. The grey zone corresponds to the desired therapeutic range and tick lines represent the lower and the upper limits of this range for the corresponding assay. Pairs in agreement, unsatisfactory and contradictory categories are symbolized by green, orange and red dots, respectively. APTT, activated partial thromboplastin time; POCT, point-of-care testing; rs, Spearman correlation coefficient.

**Figure 5 jcm-11-01338-f005:**
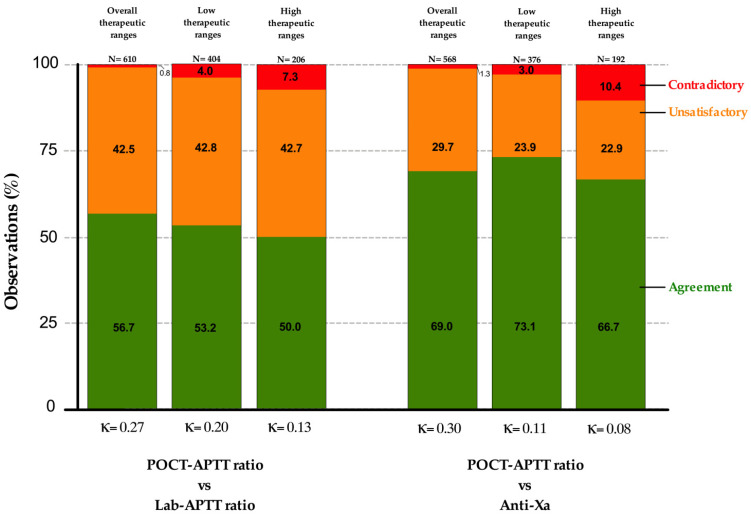
Relationships and inter-method agreements for POCT-APTT ratios vs. laboratory APTT ratios (left histograms) and POCT-APTT ratios vs. anti-Xa (right histograms), according to overall, low or high therapeutic ranges. Cohen’s Kappa coefficients are shown below each corresponding histogram.

**Figure 6 jcm-11-01338-f006:**
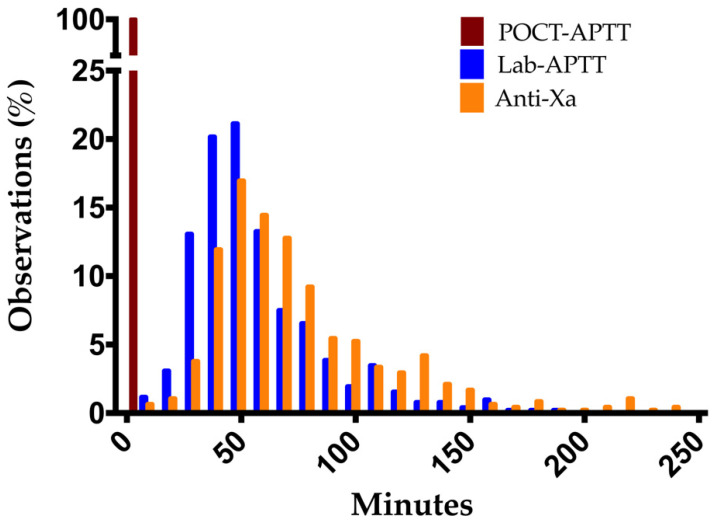
Turnaround times from sample collection to delivery results for POCT-APTT (brown), lab-APTT (blue) and anti-Xa (orange).

**Table 1 jcm-11-01338-t001:** APTT nomograms.

Heparin Therapeutic Target Range	Bleeding Risk *(Examples)*	Initiation Phase	Adjustment Phase (*aPTT Ratio*)	Comments
* **1.5–2.0 times baseline APTT** *	*Low, intermediate or* *High* (e.g., *CVVH, ECMO)*	Bolus of 25 U/kg and initial flow rate at 5 U/kg/h *	*<1.2*: bolus of 1000 U and flow rate increased by 200 U/h; check 6 h later *1.2–1.5*: flow rate increased by 100 U/h; check 4 h later *1.5–2*: the same dose is kept; check 8 h later *2–2.5*: flow rate stopped for 30 min, then decreased by 100 U/h; check 4 h later *>2.5*: flow rate stopped for 1 h and decreased by 200 U/h; check 4 h later	In order to reduce the samples collected at steady-states, coagulation checks are spaced out if the aPTT ratio is within 1.5–2.0; the next check is thus planned 8 h later if the flow rate is unchanged and 12 h later if the flow rate is unchanged on two occasions. The doses are rounded to 250 IU for boluses; to 50 units for flow rate adjustments.
* **1.5–2.0 times baseline APTT** *	Very high (e.g., *CVVH or ECMO with a recent bleeding event)*	Initial flow rate at 5 U/kg/h, except if CVVH is initiated where the initial rate is automatically 500 U/h	*<1.2*: flow rate increased by 200 U/h; check 4 h later *1.2–1.5*: flow rate increased by 100 U/h; check 4 h later *1.5–2*: the same dose is kept; check 8 h later *2–2.5*: flow rate stopped for 30 min, then decreased by 100 U/h; check 4 h later *>2.5*: flow rate stopped for 1 h and decreased by 200 U/h; check 4 h later	
* **2.0–2.5 times baseline APTT** *	Low (e.g., *fresh pulmonary embolism or DVT)*	Bolus of 80 U/kg (with a maximum of 10,000 U) and initial flow rate of 18 U/kg/h.	*<1.5*: bolus of 80 U/kg and flow rate increased by 4 U/kg/h *1.5–2*: bolus of 40 U/kg and flow rate increased by 2 U/kg/h *2–2.5*: the same dose is kept *2.5–3.0*: dose reduction of 2 U/kg/h >3: dose reduction of 3 U/kg/h, and flow rate stopped for one hour	Coagulation checks are normally done every 6 h, until equilibrium is reached. If equilibrium is reached, the next control is requested 8 h later. If two successive checks are within the aPTT ratio of 2.0–2.5, the next control is requested 12 h later.
* **2.0–2.5 times baseline APTT** *	Intermediate (e.g., *ACS, peripheral vascular ischemia, ICU patients)*	Bolus of 60 U/kg (with a maximum of 5000 U) and initial flow rate of 12 U/kg/h	*<1.5*: bolus of 60 U/kg and flow rate increased by 4 U/kg/h. *1.5–2:* bolus of 30 U/kg and flow rate increased by 2 U/kg/h *2–2.5*: the same dose is kept *2.5–3*: dose reduction of 2 U/kg/h >3: dose reduction of 3 U/kg/h, and flow rate stopped for one hour	
* **2.0–2.5 times baseline APTT** *	High (e.g., *recent postoperative patient (**≤5 days) and/or drains)*	Bolus of 40 U/kg (with a maximum of 5000 U) and initial flow rate of 12 U/kg/h.	*<1.5*: bolus of 40 U/kg and flow rate increased by 4 U/kg/h *1.5–2*: bolus of 20 U/kg and flow rate increased by 2 U/kg/h *2–2.5*: the same dose is kept *2.5–3*: dose reduction of 2 U/kg/h *>3*: dose reduction of 3 U/kg/h, and flow rate stopped for one hour	
* **2.0–2.5 times baseline APTT** *	Very high (e.g., *recent bleeding events, very recent surgery, postoperative MHV)*	No bolus.Initial flow rate of 12 U/kg/h.	*<1.5*: flow rate increased by 4 U/kg/h *1.5–2*: flow rate increased by 2 U/kg/h *2–2.5*: the same dose is kept *2.5–3*: dose reduction of 2 U/kg/h *>3*: dose reduction of 3U/kg/h, and flow rate stopped for one hour	Algorithm similar to the two previous ones except that no bolus will ever be administered neither at initiation, nor during adjustments. It is expected that the target is obtained later, but with a lower risk of exceeding it. Coagulation checks are more frequent (every 4 h instead of every 6 h) until the target is reached.

* When initiating a CVVH for which a priming of the circuit is done beforehand (10,000 U in the circuit), a start at 500 U/h is done without doing a bolus, with a coagulation control requested 4 h later. ACS, acute coronary syndrome; APTT, activated partial thromboplastin time; CVVH, continuous veno-venous hemofiltration; ICU, intensive care unit; MHV, mechanical heart valve.

**Table 2 jcm-11-01338-t002:** POCT-APTT ratio relationships with Lab-APTT ratio or anti-Xa activity.

**Lab-APTT ratio** **or** **Anti-Xa activity**	Supra-	**Disagreement** Contradictory	**Disagreement** Unsatisfactory	**Agreement** POCT =
	Therapeutic range ^a,b^	**Disagreement** Unsatisfactory	**Agreement** POCT =	**Disagreement** Unsatisfactory
	Infra-	**Agreement** POCT =	**Disagreement** Unsatisfactory	**Disagreement** Contradictory
		Infra-	Therapeutic range ^b^	Supra-
		**POCT-APTT Ratio**		

^a^: Anti-Xa therapeutic range, Low: 0.3–0.5 IU/mL, High: 0.5–0.7 IU/mL; ^b^: Lab-APTT ratio or POCT-APTT ratio therapeutic range, Low: 1.5–2.0, High: 2.0–2.5.

**Table 3 jcm-11-01338-t003:** Characteristics of the 35 patients included.

Variables	N/Median	%/IQR
**Demographics**		
Gender, females (F)	13	37.1
Age (years)	64.7	56.9–70.7
Weight (kg)	76.5	65.0–94.5
Duration of inclusion (days)	6.0	5.0–11.2
**Heparin therapeutic ranges**		
Low	23	65.7
High	8	22.9
From low to high or conversely	4	11.4
Time to desired range (hours)	29.1	15.4–37.6
**Clinical indications for UFH**		
CVVH	14	40.0
Mechanical valve	6	17.0
DVT	5	14.3
AF	3	8.6
AKI	1	2.9
ECMO	3	8.6
PE	3	8.6
**Estimated risk of bleeding**		
Low risk	2	5.7
Medium risk	16	45.7
High risk	4	11.5
Very high risk	13	37.1
**Outcome**		
Deaths in ICU	7	20.0
Major bleedings	6	17.1
Minor bleedings	6	17.1
No bleeding	23	62.9
Thrombosis	0	0.0

Categorical variables expressed as number (%); continuous variables as median (IQR). AF, atrial fibrillation or flutter; AKI, acute kidney injury; CVVH, continuous veno-venous hemofiltration; DVT, deep vein thrombosis; ECMO, extracorporeal membrane oxygenation; PE, pulmonary embolism; UFH, unfractionated heparin.

## Data Availability

Please contact the corresponding author.

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
