# Peer review of "Monitoring of Unfractionated Heparin Therapy in the Intensive Care Unit Using a Point-of-Care aPTT: A Comparative, Longitudinal Observational Study with Laboratory-Based aPTT and Anti-Xa Activity Measurement"

_jcm, 2022, doi:10.3390/jcm11051338_

Round 1

Reviewer 1 Report

Dear colleagues,

Thank you for this nice manuscript. Anticoagulation in the intensive care unit is an everyday topic and of high relevance for man clinicians. The selection of the anticoagulant as well as the selection of the adequate monitoring option is always the subject of discussions. Therefore, your work addresses an important issue.

I have a few questions after reading it:

  1. How was the amount of cases/number of samples calculated?
  2. Why did you include patients on ECMO or CVVHD? It is well-known that extracorporeal devices influence platelets, platelet function and interferes with especially anti-Xa monitoring. Choosing the right reagent is important here. There are large differences in the correlation between the individual aPTT- and anti-Xa tests. This could certainly have led to a poorer correlation of the samples. These are difficult collectives, and a more homogeneous collective would certainly have been better for such a topic. Maybe you could re-analyze the results without ECMO and CVVHD patients?!
  3. You could address the problems of ECMO influence more precisely on anti-Xa measurement.
  4. What are the differences to the studies that advocate monitoring with anti-Xa? Why did this work better there?

Author Response

We would like to thank the JCM editors and reviewers for their thoughtful and detailed comments on our paper. Please find the specific authors’ responses to the reviewers below. (Please see the file in attachment).

Dear colleagues,

Thank you for this nice manuscript. Anticoagulation in the intensive care unit is an everyday topic and of high relevance for man clinicians. The selection of the anticoagulant as well as the selection of the adequate monitoring option is always the subject of discussions. Therefore, your work addresses an important issue.

  • Authors’ response: Thank you for your comment.

I have a few questions after reading it:

1. How was the amount of cases/number of samples calculated?

  • Authors’ response: The number of patients was chosen arbitrarily. The research was exploratory and therefore no power and sample size estimation was performed. This has been added in the manuscript in the Methods part as follows:

Line 108: “Due to the exploratory design of the study, no formal sample size calculation could be performed.”

2. Why did you include patients on ECMO or CVVHD? It is well-known that extracorporeal devices influence platelets, platelet function and interferes with especially anti-Xa monitoring. Choosing the right reagent is important here. There are large differences in the correlation between the individual aPTT- and anti-Xa tests. This could certainly have led to a poorer correlation of the samples. These are difficult collectives, and a more homogeneous collective would certainly have been better for such a topic. Maybe you could re-analyze the results without ECMO and CVVHD patients?!

  • Authors’ response: Thank you for this highly relevant comment. As suggested, we re-analyzed the results for 18 patients excluding ECMO and CVVH patients (n=17). The results are shown below (Graph1 & Graph 2). As you suggested, correlations (Spearman rs) observed are higher in this subsample than in the whole cohort, particularly for the POCT-APTT vs Anti-Xa comparison (0.59 vs. 0.46) and also when tests results are expressed as ratios, particularly for the POCT-APTT ratio vs Anti-Xa comparison (0.58 vs 0.50).

Graph 1 : Overall correlation of POCT-APTT vs laboratory APTT (left panel, N=401) and POCT-APTT vs Anti-Xa (right panel, N=366) from the 18 patients without ECMO and CVVH.

Graph 2: Correlations of POCT-APTT ratios vs laboratory APTT ratios (left panel) and POCT-APTT ratios vs chromogenic Anti-Xa activity measurement (right panel) according to overall therapeutic target ranges.

This poorer correlation of the samples observed has been nuanced in the manuscript in the  “Results” and “Discussion” parts. References have also been added as follows:

Line 271 (Results): "After excluding the 17 patients on ECMO and CVVH (n=18), we observed the following rs for the POCT-APTT vs lab-APTT and POCT-APTT vs Anti-Xa: 0.79 (95% CI, 0.75-0.82, p<0.0001) and 0.59 (95% CI, 0.51-0.65, p<0.0001), respectively."

Line 416 (Discussion): “(…)Since 48.6% of the patients included in this study were on ECMO (8.6%) or CVVH (40.0%), this led to a poorer correlation between POCT-APTT and Anti-Xa pairs.”

3. You could address the problems of ECMO influence more precisely on anti-Xa measurement.

  • Authors’ response: Thank you for your comment. We had already addressed the analytical interferences encountered with Anti-Xa chromogenic assay but we had not precise this could be particularly encountered in patients on ECMO. We therefore added (and coupled with the reviewer1’s comment 2) in the discussion part of the manuscript as follows:

Line 404: “According to two previous studies, increased plasma free hemoglobin concentration resulted in a concentration-dependent underestimation of heparin activity [29,30]. As a consequence of shear stress and exposure of blood to non-biological substance in ECMO patients, thrombocytopenia and altered platelet function may occur due to decreased levels of adhesion receptors, activation markers and surface expression of CD62/CD63 [9,31]. This may also underestimate UFH effects given measurement of Anti-Xa activity in a platelet-free environment. Some patients may also experience high levels of fibrinogen during the ECMO course, to which anti-Xa assays are not sensitive. All of these conditions frequently encountered in patients on extracorporeal devices may therefore lead to inadequate UFH dosing[29–32]. (…)”

4. What are the differences to the studies that advocate monitoring with anti-Xa? Why did this work better there?

  • Authors’ response: Although recent studies propose the anti-factor Xa to be the gold standard for monitoring UFH anticoagulation, especially in ECMO patients, this strategy is associated with potentially long intervals before dose adjustments, thus increasing the time required to obtain the therapeutic heparin range which could be detrimental to the patient. The best strategy for monitoring UFH anticoagulation could not be determinated here as the aims of the study consisted in a comparison of this POCT version of the APTT to laboratory-based APTT and measurement of anti-Xa activity in terms of correlation, agreement, relationship and turnaround time (TAT) in patients under UFH therapy in the ICU. Further large prospective studies could assess whether a monitoring method is associated with a lower incidence of adverse outcomes, i.e. thrombosis and haemorrhages.

Reviewer 2 Report

This is a very interesting study in a very important, yet largely neglected area of hemostasis research. The study is sound and the study shortcomings are apprpiately adressed by the authors. If have still some concerns that should be taken into account:

1) The correlation between lab-aPTT and anti-Xa is not presented in the manuscript but shall be of interest to readers.

2) When assessing confounding factors, the authors missed out F12 and F11 deficiency which is frequently seen in patients with extracorporal circulation such as, ECMO. In our hands, deficiency of contact factors - which develops over the first few days - is a major driver of discrepancy between anti-Xa assays and aPTT and also dangerous. I assume that the authors do not have access to any spare samples to analyse F12/F11 but they should discuss this point. If the cohorts are not getting too small, they may also show statistically whether the difference between anti-Xa and aPTTs (both lab and POCT) is larger in patients with assist devices than in those without.

Author Response

We would like to thank the JCM editors and reviewers for their thoughtful and detailed comments on our paper. Please find the specific authors’ responses to the reviewers below.

This is a very interesting study in a very important, yet largely neglected area of hemostasis research. The study is sound and the study shortcomings are apprpiately adressed by the authors. If have still some concerns that should be taken into account:

  • The correlation between lab-aPTT and anti-Xa is not presented in the manuscript but shall be of interest to readers.
  • Authors’ response: We agree with the reviewer that this is also an interesting topic. However, the objective of the study was the evaluation of the POCT device. To avoid multiple statistical tests and analyses, and thus the risk of type I error, we prefer to avoid the comparison of laboratory APTT with anti-Xa. This could be the subject of another study, though.

  • When assessing confounding factors, the authors missed out F12 and F11 deficiency which is frequently seen in patients with extracorporal circulation such as, ECMO. In our hands, deficiency of contact factors - which develops over the first few days - is a major driver of discrepancy between anti-Xa assays and aPTT and also dangerous. I assume that the authors do not have access to any spare samples to analyse F12/F11 but they should discuss this point. If the cohorts are not getting too small, they may also show statistically whether the difference between anti-Xa and aPTTs (both lab and POCT) is larger in patients with assist devices than in those without.
  • Authors’ response: We thank the reviewer for this interesting comment. We were able to retrieve an average of 1 sample every 72 hours in the majority of study patients. In our hands, the levels of factors XI and XII were also important determinants of both lab-APTT and POCT-APTT, and also explained differences between the two tests (Table S1, see the attachment). The addition of these variables also provided a more complete multivariable model (Table S2, see the attachment). Changes have been made in the text to add these elements:

Line 184 (Methods): The following additional laboratory tests were performed in parallel with the POCT-APTT, lab-APTT and anti-Xa assays: antithrombin (AT; Liatest ATIII reagent, Stago), C-reactive protein (CRP; CRP Gold Latex®, DiAgam, Ghislenghien, Belgium), fibrinogen (Fg; STA Liquid Fib reagent, Stago), factor VIII (FVIII; STA Deficient VIII and STA-C.K. Prest reagents, Stago), factors XI and XII (FXI & FXII; STA Immunodef XI, STA Immunodef XII and STA-C.K. Prest reagents, Stago) and LA.

Line 452 (Discussion): “We also assessed the variables that could explain differences between POCT-APTT and lab-APTT and found that CRP, fibrinogen, factor XII levels, and LA positivity could explain differences between the two tests. Recent studies found that FXII and FXIII shows a significant decrease during ECMO), FXIII even showing reduced levels hours before its initiation [39,40]. Low coagulation FXII activity had been associated with less thromboembolic complications [40]and was beneficial in an animal model as its inhibition prevented fibrin deposition and thrombosis in the extracorporeal circuit [41]. As the whole cohort in our study included patients under extracorporeal circuit, we assumed it could influence the correlations between POCT-APTT vs Anti-Xa assay and POCT-APTT vs lab-APTT.

Line 483 (Discussion) : “Although we studied the effect of known confounding factors on APTT measurements and found that CRP, fibrinogen, factor XII levels and LA positivity were the principal factors that could modulate the relationship between POCT- and lab-APTT, there was a lot of between- and within-patient variability that was not related to UFH levels and that we could not explain with our model [38]”

Reviewer 3 Report

This is a well-written paper on the comparison of point-of-care aPTT comparison with lab-based aPTT and anti-Xa in an ICU

Abstract: leave out mentioning of citrated plasma. This belongs to Methods

Introduction: expand on the aim; e.g., accuracy? Turnaround time?

Methods: thorough and fine

Results: Figure 3 is difficult to read. Make each ‘patient’ table bigger

Discussion: well balanced.

Author Response

We would like to thank the JCM editors and reviewers for their thoughtful and detailed comments on our paper. Please find the specific authors’ responses to the reviewers below.

This is a well-written paper on the comparison of point-of-care aPTT comparison with lab-based aPTT and anti-Xa in an ICU

- Abstract: leave out mentioning of citrated plasma. This belongs to Methods.

  • Authors’ response: Thank you for your comment. The sentence has been modified and “109mM buffered citrated plasma” has been deleted.

- Introduction: expand on the aim; e.g., accuracy? Turnaround time?

  • Authors’ response: Thank you for this relevant remark. We have moved part of the first paragraph of "methods" in the introduction in order to expand the aim in the introduction section as follows: 

 Line 81 : “The aim of the study was to compare this POCT version of the APTT to laboratory-based APTT and measurement of anti-Xa activity in terms of correlation, agreement, relationship and turnaround time (TAT) in patients under UFH therapy in the ICU.

- Methods: thorough and fine

  • Authors’ response: Thanks.

- Results: Figure 3 is difficult to read. Make each ‘patient’ table bigger.

  • Authors’ response: We fully agree with the reviewer. We have modified Figure 3 by displaying it in landscape mode to better appreciate the details. Therefore each « patient graph » now appears bigger.

- Discussion: well balanced.

  • Authors’ response: Many thanks.